# Molecular mechanism of gallium nitrate in inhibiting bacterial biofilm formation through *pykF* modulation

**Xiaofeng Zhang, Junjie Dong*****, Bing Wang, Lingqiang Chen, Zhiqiang Gong, Jin Yang*, Guizhao Shu, Qi Ning**

The First Affiliated Hospital of Kunming Medical University, Kunming, China

* kmdjj1223@163.com (JD); 554879537@qq.com (JY)

## Abstract

### Purpose

Gallium nitrate, a non-redox analog of iron (III), suppresses bacterial biofilms and virulence within the framework of bacterial regulation. This study investigates the molecular mechanisms and regulatory pathways through which gallium nitrate modulates bacterial activity and function.

### Methods

The antimicrobial properties of gallium nitrate, its effects on bacterial biofilms, and gallium-responsive signaling pathways were assessed. Observation of marked upregulation of pyruvate kinase (*pykF*) expression following gallium nitrate exposure prompted *in vitro* and *in vivo* experiments to examine how gallium influences the expression, enzymatic activity, and functional role of bacterial *pykF*.

### Results

Crystal violet staining, XTT assay, confocal laser scanning microscopy, and scanning electron microscopy consistently indicated that gallium nitrate suppressed bacterial biofilm formation and metabolic activity. Transcriptomic profiling and subsequent validation analyses further suggested a strong association between *pykF* and gallium-mediated antibacterial effects. Both *in vitro* and *in vivo* experiments revealed that *pykF* knockout significantly enhanced bacterial survival and biofilm formation.

### Conclusion

Gallium nitrate modulates bacterial biofilm development and virulence, with its antimicrobial effect largely dependent on *pykF* upregulation. Concurrent therapeutic targeting of both *pykF* and gallium may provide a more effective strategy against persistent biofilm-associated infections. This work also establishes a mechanistic basis

**Data availability statement:** All data relevant to the manuscript have been provided and are available as unrestricted open access. Experimental data were included in the "Supporting Information" and "Original WB data and molecular weight markers in the article" upon manuscript submission. Sequencing data in the manuscript have been deposited in the Mendeley Data repository (URL: https://data.mendeley.com/). The DOI is: Zhang, Xiaofeng (2025), "Organize raw data according to the images in the article", Mendeley Data, V2, doi: 10.17632/wrs5sx4bhb.2. Direct access to the data is available via the provided link.

**Funding:** the Major Science and Technology Project of Yunnan Provincial Department of Science and Technology, Yunnan Provincial Orthopedic and Sports Rehabilitation Clinical Medicine Research Center Yunnan Provincial Endocrinology and Metabolism Clinical Medicine Center Yunnan Provincial Endocrinology and Metabolism Clinical Medicine Center Master's Innovation Fund of the First Affiliated Hospital of Kunming Medical University.

**Competing interests:** The authors have declared that no competing interests exist.

for clinical approaches aimed at reducing biofilm formation and limiting device-related infections.

---

Intramedullary implants are indispensable in bone and joint repair and reconstruction, yet infections associated with their use often trigger acute or chronic osteomyelitis. Such infections can cause bone defects and nonunion, with risks of surgical failure and substantially elevated treatment costs. Postoperative implant infection rates in orthopedic surgery are estimated at approximately 5% [1]. Current therapeutic approaches mainly rely on surgical debridement combined with antimicrobial therapy [2]. Nonetheless, the persistence of bacterial biofilms—reinforced by extracellular polymeric substances, localized microenvironments, altered expression of biofilm-related genes, and secretion of antibiotic-hydrolyzing enzymes [3]—remains a major clinical barrier. Conventional antimicrobial agents typically penetrate biofilms poorly, limiting their capacity to eliminate intramedullary implant-related infections. Consequently, the development of innovative strategies and materials capable of ensuring durable control of intramedullary implant infections is an urgent research priority.

Gallium ions and their compounds have been implicated in infection control, anti-cancer therapy, and the regulation of bone metabolism [4]. Their antimicrobial activity is mediated through inhibition of bacterial biofilm formation, achieved by disrupting iron metabolism, acting as antibiotic adjuvants, or forming complexes such as gallium porphyrins [5–8]. Co-crystallization of proflavine with metal-based agents, including silver, copper, zinc, and gallium, has been shown to produce potent antimicrobial effects at concentrations below 0.125 mg/mL [9]. Evidence indicates that inhibition of Streptococcus mutans via the *PykF* target was identified in studies conducted by researchers other than Han Wang and colleagues, and that lentinan was responsible for this activity [10]. Furthermore, surfactin-conjugated silver nanoparticles have been reported to exert strong antimicrobial and antibiofilm effects, effectively targeting resistant pathogens such as *Pseudomonas aeruginosa* [11]. Gallium nitrate-coated $TiO_2$ nanotubes were also demonstrated to markedly reduce the development of mixed biofilms composed of *Staphylococcus aureus* and *Escherichia coli* [12].

Pyruvate kinase, encoded by the *pykF* gene, suppresses bacterial proliferation through pathways such as the Vvrr1–*pykF* axis and plays a central role in energy metabolism [13,14]. Based on these insights, transcriptomic sequencing was applied in the present study to examine whether gallium nitrate impedes bacterial biofilm formation by modulating *pykF* mRNA expression.

*In vitro* and *in vivo* analyses were conducted to delineate the antimicrobial mechanisms of gallium nitrate from two dimensions: (1) functional phenotypes and omics, focusing on its antimicrobial activity, modulation of bacterial biofilms, and influence on gallium-related bacterial signaling pathways; and (2) molecular mechanisms, addressing the inhibition of bacterial growth through regulation of *pykF* mRNA expression and the potential roles and modes of action of pyruvate kinase. Clarifying the mechanistic basis of gallium and its derivatives establishes both an experimental

foundation and a theoretical framework for the advancement of novel gallium-based antimicrobial agents and surface coatings.

## 1. Results and discussion

### 1.1 Gallium nitrate possesses good antimicrobial properties and inhibits bacterial biofilm formation

The bacteriostatic activity of gallium nitrate was assessed by exposing bacteria to varying concentrations of the compound. The MIC for *E. coli*, *S. aureus*, and *P. aeruginosa* exceeded 1024 µg/mL, as bacterial proliferation persisted at this concentration (S1 Fig).

Crystal violet staining was subsequently employed to evaluate its effect on biofilm formation. Relative to the control, biofilms produced by *E. coli*, *S. aureus*, and *P. aeruginosa* exhibited marked reduction in biomass following gallium nitrate exposure, indicating inhibition of biofilm development (Fig 1A).

SEM revealed substantial alterations in biofilm structure, while CLSM demonstrated diminished bacterial viability and decreased biofilm thickness. Treatment with gallium nitrate sharply reduced the bacterial load within the biofilms of all three species (Fig 1B-a). Moreover, the surface morphology of the bacteria became notably irregular, with disrupted biofilm integrity and abundant cell debris (Fig 1B-b). Dual staining with SYTO9 and PI confirmed a reduction in viable biofilm biomass after treatment (Fig 1C), accompanied by a pronounced decline in the T/C ratio (Fig 1D), reflecting a greater proportion of non-viable cells within the biofilms. Three-dimensional reconstruction further showed a significant decrease in biofilm thickness for *S. aureus*, whereas reductions in *E. coli* and *P. aeruginosa* were not statistically significant (Fig 1D). Taken together, the results indicate that gallium nitrate not only suppresses bacterial viability but also impedes biofilm formation.

### 1.2 *PykF* as a potential key target for gallium nitrate inhibition of bacteria

Prokaryotic transcriptome sequencing was conducted on control and gallium nitrate–treated *E. coli* strains to clarify the mechanisms underlying gallium nitrate–mediated inhibition of biofilm formation and antibacterial activity. Analysis identified 2020 significantly differentially expressed genes, including 1030 upregulated and 990 downregulated transcripts. A heatmap illustrating gene expression clustering is presented in Fig 2A. Differentially expressed genes were subjected to Kyoto Encyclopedia of Genes and Genomes (KEGG) enrichment analysis via the KEGG Orthology Based Annotation System. The top 20 pathways ranked by statistical significance were visualized in a bubble plot using ggplot2 (Fig 2B), highlighting marked alterations in amino acid biosynthesis, pyruvate metabolism, and carbon metabolism. A Venn diagram comparing the three significantly enriched pathways (Fig 2C) revealed *pykF* and *pykA* as commonly upregulated genes (Fig 2D). To validate the impact of gallium nitrate on *pykF* and *pykA* expression, the sample size was expanded and qPCR was employed. Gallium nitrate treatment significantly increased *pykF* mRNA expression, whereas *pykA* expression changes were not statistically significant (Fig 3A). Further confirmation of *pykF* upregulation was obtained by assessing protein levels in three bacterial species exposed to gallium nitrate using WB, which consistently demonstrated elevated *pykF* expression across all strains (Fig 3B).

In vivo and *in vitro* evidence indicates that *pykF* modulates bacterial growth and biofilm formation. To further validate the influence of *pykF* on bacterial physiology, analyses were performed at both the molecular binding and gene pathway levels. The protein structure of *pykF* (1E0T) was retrieved from the RESC.PDB [1] database, and the 3D structure of Ga metal ions was obtained. Molecular docking was carried out using AutoDock4.2.6 [4], with the docking grid centered on the principal structural domain of the protein and a grid spacing of 0.55 Å. All potential binding regions were included, while default values were applied for the remaining parameters. The most favorable binding conformations were determined, and the 50 highest-scoring structures from each docking run were selected for evaluation. The analysis demonstrated that Ga ions interacted with multiple sites on *pykF*, exhibiting a lowest binding energy of −6.91 kcal/mol.

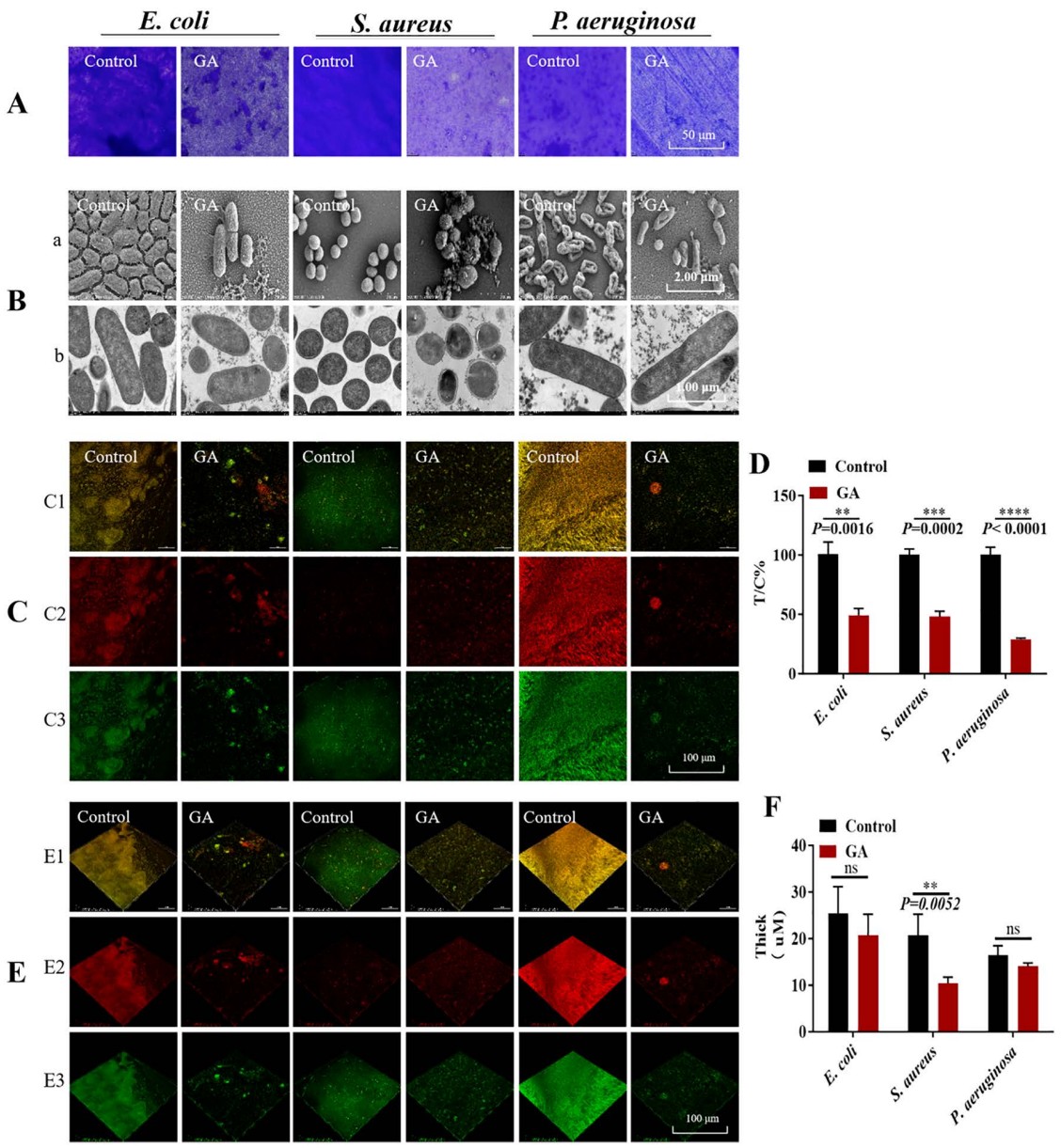

**Fig 1. Antibacterial activity of gallium nitrate and its inhibition of bacterial biofilm formation.** A: Crystal violet staining assessing the influence of gallium nitrate on biofilm formation, with 0 μg/mL in the control group and 1024 μg/mL in the gallium nitrate treatment group (GA). B: Ultrastructural alterations in bacterial biofilm. a: Scanning electron microscopy images of biofilm morphology. b: Transmission electron microscopy images highlighting structural modifications. C: Confocal laser scanning microscopy analysis of biofilm formation with an argon ion laser as the excitation source. C1: Fluorescent costaining of live and dead bacteria. C2: Red fluorescence indicating dead bacteria. C3: Green fluorescence indicating viable bacteria. D: XTT assay evaluating the effect of gallium nitrate on cell viability dynamics within the biofilm (live-to-dead cell ratio). E: Three-dimensional reconstruction of biofilm thickness. E1: Fluorescent costaining of live and dead bacteria. E2: Red fluorescence indicating nonviable cells. E3: Green fluorescence indicating viable cells. F: Quantitative analysis of biofilm thickness based on the proportion of live (green) and dead (red) cells, with the number of layers recorded and subjected to statistical evaluation. *P < 0.05, **P < 0.01, ***P < 0.001, ****P < 0.0001.

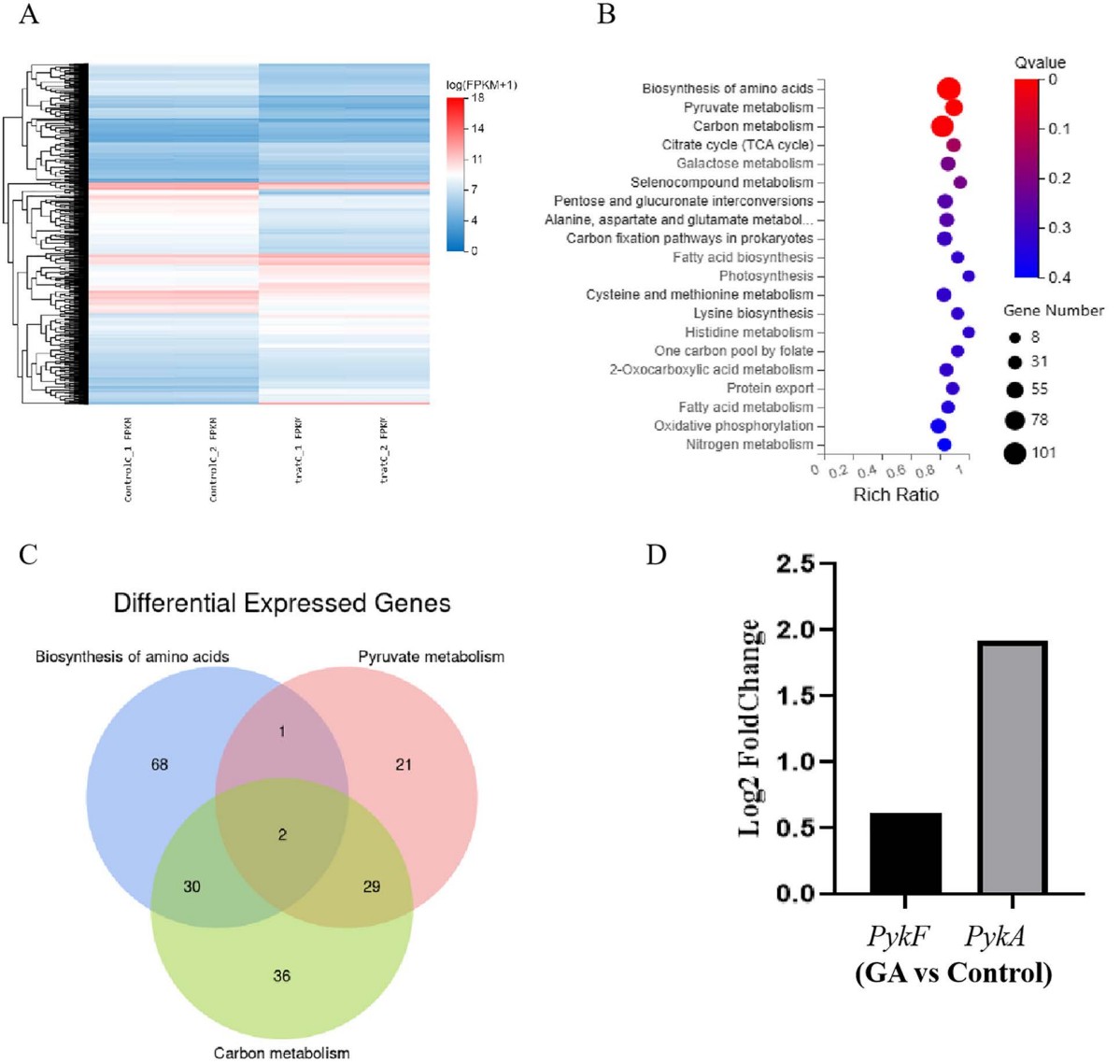

**Fig 2. Transcriptome sequencing analysis of gallium nitrate–treated bacteria demonstrates suppression of biofilm formation.** A: Heatmap illustrating differentially expressed genes, with blue indicating reduced expression and red indicating elevated expression. B: KEGG pathway enrichment analysis of differentially expressed genes, where the x-axis denotes the proportion of annotated genes within each pathway relative to the total, and the y-axis lists the KEGG pathways. Dot size corresponds to the number of genes, and the color gradient from blue to red reflects enrichment significance. C: Venn diagram showing the overlap of genes within significantly enriched KEGG pathways. D: Bar plot presenting expression profiles of selected differentially expressed genes.

Specifically, Ga formed two coordination bonds with the side chain carboxyl group of E222 at distances of 2.9 and 3.1 Å, and two additional coordination bonds with residue D246 at 2.9 and 3.8 Å (Fig 4A). These interactions indicate that Ga ions bind effectively to *pykF*. To complement structural analysis, single-gene *GSEA* was performed for *pykF*, identifying 48 significantly enriched pathways (|NES| > 1, p < 0.05, q < 0.25), many of which were associated with energy metabolism and amino acid biosynthesis. Venn analysis of the identified pathways and those previously associated with bacterial growth and biofilm development revealed five overlapping pathways (Fig 4B): phosphotransferase system (pts), nitrogen

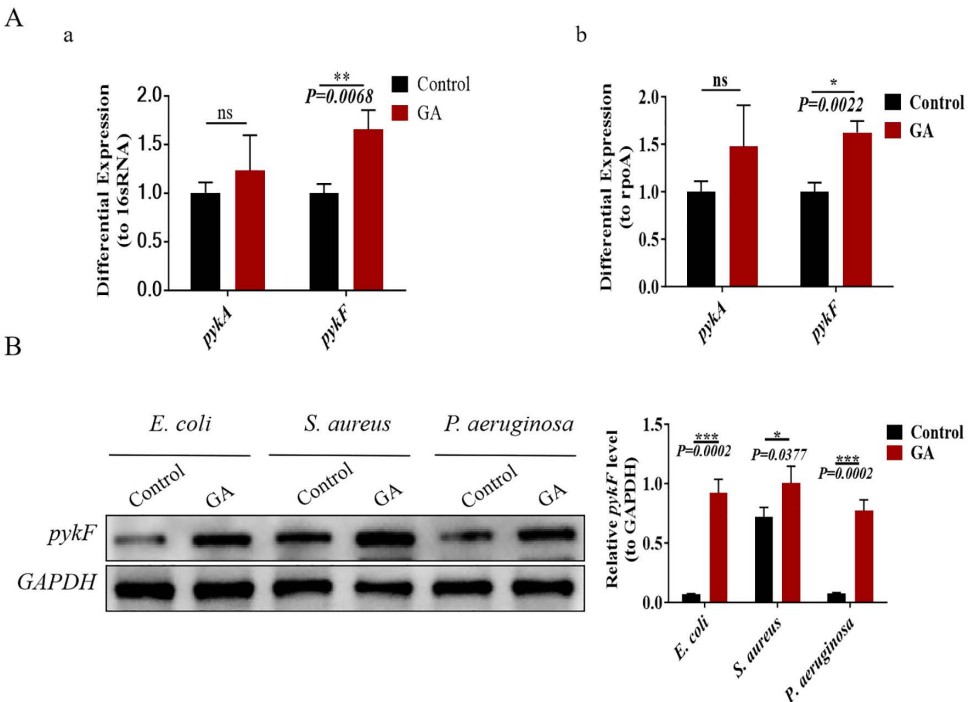

**Fig 3. Experimental confirmation of differential gene expression.** A: a. Reverse transcription polymerase chain reaction (RT-PCR) quantification of *pykA* and *pykF* mRNA levels in *E. coli*, normalized to 16sRNA. Relative to the control group, *pykA* expression in the GA group increased by 1.233-fold, and *pykF* by 1.657-fold. b. RT-PCR quantification of *pykA* and *pykF* using *rpoA* as the internal control revealed 1.472-fold and 1.623-fold increases, respectively, in the GA group compared to the control. B: Western blot detection of *pykF* protein levels in *E. coli*, *S. aureus*, and *P. aeruginosa*, with GAPDH as the internal reference. Relative fold changes in the GA group compared with controls were 13.056 for *E. coli*, 1.395 for *S. aureus*, and 10.349 for *P. aeruginosa*. *P<0.05, **P<0.01, ***P<0.001, ****P<0.0001.

metabolism, amino sugar and nucleotide sugar metabolism, carbon metabolism, and oxidative phosphorylation. All five pathways exhibited positive correlations with *pykF* enrichment (Fig 4C), supporting a central role of *pykF* in regulating bacterial proliferation and biofilm formation. Genes mapped to these pathways displayed marked upregulation following Ga metal exposure (Fig 4D), implying that Ga ions enhance the transcription of these genes and, in turn, restrict bacterial growth and biofilm establishment. Expression profiling of genes with the highest differential fold changes across pathways demonstrated a marked reduction in expression following *pykF* knockout (Fig 4E). Consistently, the knockout strain exhibited a significantly slower growth rate compared with the wild-type strain (Fig 4F), supporting the functional relevance of *pykF* in bacterial proliferation. Growth patterns also varied according to inoculation density after 12 hours in the logarithmic phase, with maximal fitness observed at an OD600 of 0.1 (Fig 4G). Under exposure to graded concentrations of gallium nitrate, the knockout strain displayed greater tolerance relative to the wild type. At 64 μM, growth of both strains was suppressed to comparably low levels (Fig 4H), indicating that *pykF* serves as a primary target of gallium nitrate–mediated antibacterial activity. Molecular docking further suggested that $Ga^{3+}$ bound to the *pykF* protein, potentially altering its structural integrity or catalytic capacity. Transcriptomic and proteomic analyses showed a pronounced upregulation of *pykF* mRNA and protein expression under $Ga^{3+}$ treatment, implying an indirect regulatory influence through stress-adaptive or metabolic signaling pathways. No evidence currently supports direct interaction between $Ga^{3+}$ and the *pykF* gene at the DNA level; the observed transcriptional activation is more plausibly attributed to gallium-induced metabolic stress or perturbation of iron-dependent transcriptional regulators. Clarification of these upstream mechanisms will require additional investigations, including promoter activity assays and chromatin immunoprecipitation (ChIP).

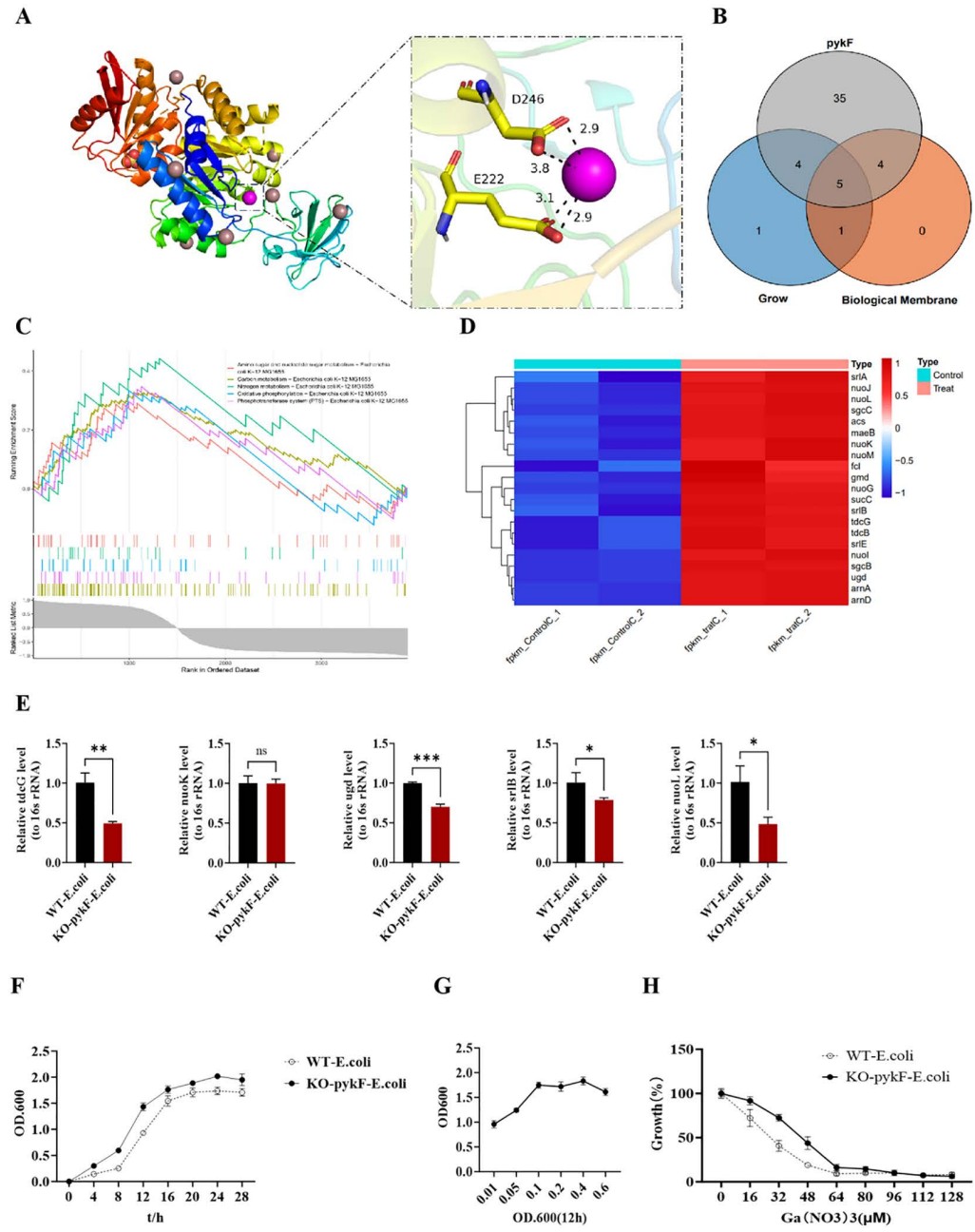

**Fig 4. Ga³⁺ binding to *pykF* and its impact on *E. coli* growth.** A: Molecular docking analysis of Ga³⁺ with *pykF*, indicating multiple binding sites, with the purple site representing the maximum absolute binding energy. B: Venn analysis of single-gene *GSEA* pathways associated with *pykF*, highlighting links to bacterial growth and biofilm formation. C: Pathways enriched in *pykF* related to bacterial growth and biofilm formation. D: Heatmap of gene expression in *pykF*-enriched pathways connected to bacterial growth and biofilm formation. E: Comparative expression of pathway genes in wild-type and *pykF* knockout strains. F: Growth curves of wild-type versus *pykF* knockout strains. G: Fitness profiles of *pykF* knockout strains. H: Comparative tolerance of wild-type and *pykF* knockout strains to gallium nitrate.

Molecular docking confirmed a strong affinity of Ga³⁺ for the *pykF* protein; however, the biological implications of this interaction remain unverified. As *pykF* (pyruvate kinase I) functions in the terminal stage of glycolysis by catalyzing the conversion of phosphoenolpyruvate (PEP) to pyruvate with concomitant ATP generation, structural alterations induced by metal ion binding may plausibly influence enzymatic activity. Despite the absence of canonical iron-binding motifs such as Fe-S clusters, *pykF* typically requires divalent cofactors, including $Mg^{2+}$ or $K^+$, to sustain catalytic function. The extent to which Ga³⁺ interacts with these sites, acting as either an inhibitor or activator, has not been determined. Clarification of whether Ga³⁺ modulates *pykF* enzymatic activity directly or primarily regulates its transcriptional and translational levels requires additional kinetic and structural investigations.

### 1.3 Effect of gallium nitrate on *pykF*-knockout strains

**1.3.1 *In vitro* study on gallium nitrate's *pykF mRNA* upregulation and biofilm inhibition.** To validate the role of gallium nitrate in suppressing biofilm formation via regulation of bacterial *pykF* mRNA, *pykF*-deficient *E. coli* strains were generated (knockout strategy illustrated in S2A Fig). qPCR and WB analyses were then performed to evaluate the influence of gallium nitrate and *pykF* deletion on *pykF* expression. Loss of *pykF* markedly reduced mRNA levels ($p < 0.05$). Gallium nitrate treatment induced an upward trend in *pykF* mRNA, and comparison between the KO-*pykF* group and the *KO-pykF* + GA group revealed higher transcript abundance in the latter, although the difference was not statistically significant (S2B-a Fig). At the protein level, however, gallium nitrate markedly enhanced PykF expression (S2B-b Fig), suggesting transcriptional activation followed by translation as a central mechanism underlying the inhibition of biofilm development.

Interestingly, in Ga-treated KO-*pykF* strains, mRNA levels remained largely unchanged, while protein levels increased substantially, indicating a divergence between transcription and translation. This outcome implies a possible post-transcriptional regulatory process. Ga³⁺ may promote ribosome engagement or initiation of translation by modulating mRNA secondary structure or by improving ribosomal efficiency under stress. Another possibility is that Ga³⁺ stabilizes residual *PykF* protein or enables translation from truncated or cryptic transcripts. Verification of these possibilities requires further exploration through ribosome profiling, RNA structural interrogation, and protease degradation assays.

To clarify the role of *pykF* in mediating the effects of gallium nitrate on bacterial activity and biofilm formation, *pykF*-KO *E. coli* and the control strain were subjected to gallium nitrate treatment. CLSM was applied to evaluate bacterial activity and biofilm thickness (Fig 5A–5D), and SEM was used to examine biofilm ultrastructure (Fig 5E). The biofilm formed by the *pykF*-deficient strain contained markedly higher bacterial density compared with the control. CLSM analysis revealed a significant elevation in the T/C ratio after *pykF* deletion, indicating that pyruvate kinase inhibits biofilm development. Three-dimensional reconstruction based on SYTO9 and PI staining further demonstrated a significant increase in biofilm thickness following *pykF* knockout, whereas SEM analysis showed minimal alteration in biofilm ultrastructure.

Subsequent treatment of both *pykF*-deficient and control strains with gallium nitrate revealed marked changes. SEM demonstrated roughened bacterial surfaces, disrupted biofilm connectivity, and abundant cellular fragments. CLSM confirmed a reduction in bacterial viability, reflected by a significant decrease in the T/C ratio, alongside thinner biofilms. Compared with the KO-*pykF* group, the KO-*pykF* + GA group exhibited a pronounced decline in bacterial abundance within biofilms. Collectively, the results indicate that gallium nitrate suppressed biofilm formation and bacterial viability in both wild-type and *pykF*-deficient *E. coli*. Further comparison between the *KO-pykF* + GA group and the *KO-NC* + GA group demonstrated a marked elevation in the abundance of *E. coli* within the biofilm of the *KO-pykF* + GA group. CLSM analysis revealed a significant increase in the T/C ratio accompanied by partial recovery of biofilm thickness. SEM examination showed that bacterial morphology remained largely intact, intercellular connectivity was enhanced, and cellular debris was reduced. Collectively, these observations indicate that gallium nitrate exerted comparable inhibitory effects on bio-film formation in both wild-type and *pykF*-deficient strains. Furthermore, under conditions of *pykF* deletion, gallium nitrate partially restored biofilm structure, implying a regulatory association between *pykF* expression and biofilm development in

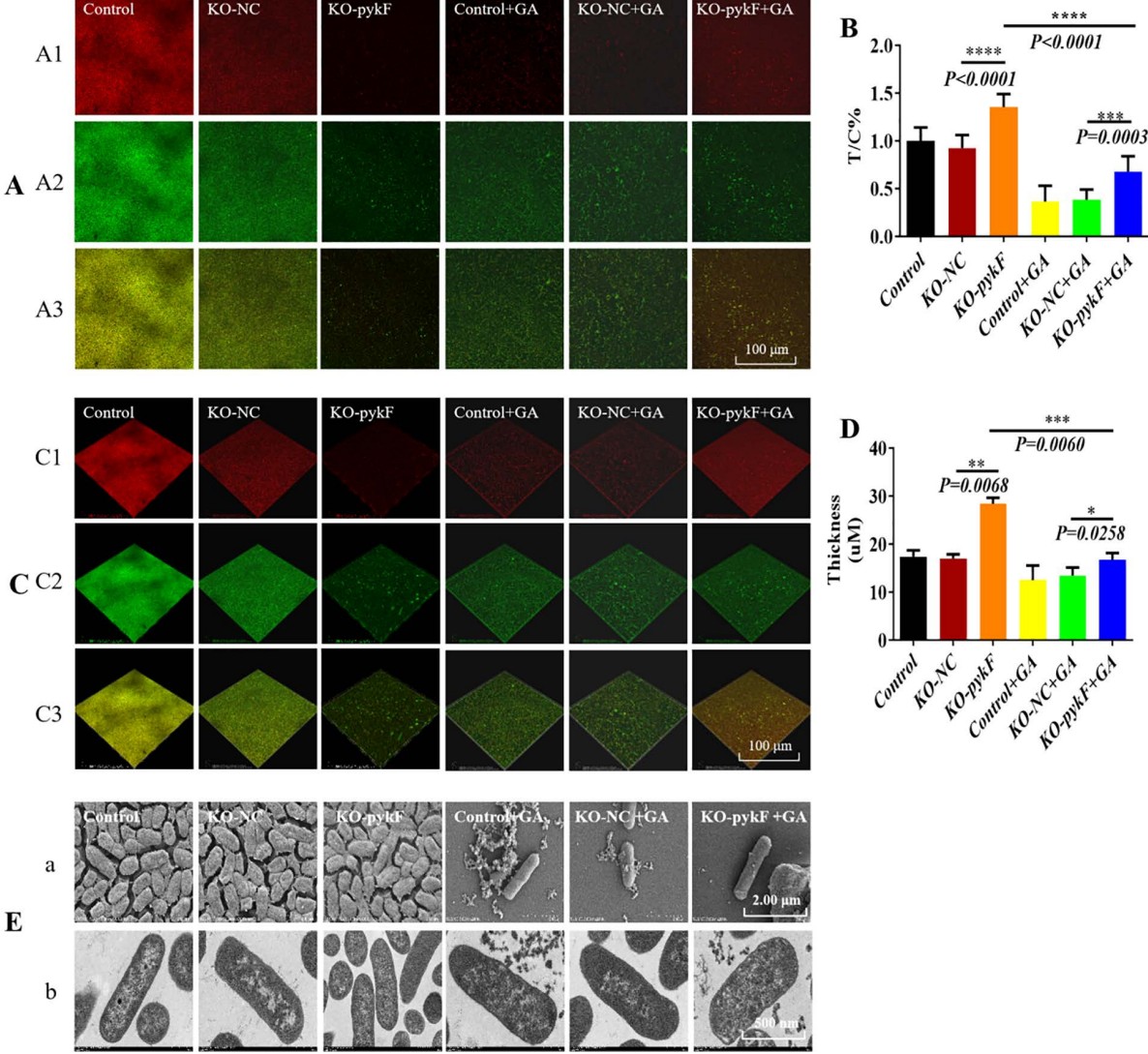

**Fig 5. Combined impact of gallium nitrate exposure and *pykF*-knockout on biofilm thickness, formation capacity, and ultrastructural alterations.** A: Confocal laser scanning microscopy assessment of biofilm formation capacity. A1: Dead bacteria exhibiting red fluorescence. A2: Live bacteria exhibiting green fluorescence. A3: Fluorescent costaining of live and dead bacteria. B: XTT assay evaluating dynamic changes in biofilm development, expressed as T/C. C: Three-dimensional reconstruction of biofilm thickness. C1: Red fluorescence indicating nonviable bacteria. C2: Green fluorescence indicating viable bacteria. C3: Fluorescent costaining of live and dead bacteria. D: Quantitative analysis of biofilm layers based on the distribution of viable (green) and nonviable (red) cells. E-a: Scanning electron microscopy visualization of ultrastructural changes (scale bar: 2.0 μm). E-b: Transmission electron microscopy visualization of ultrastructural changes (scale bar: 500 nm). *$P < 0.05$, **$P < 0.01$, ***$P < 0.001$, ****$P < 0.0001$.

the presence of gallium nitrate. Overall, the data demonstrate that inhibition of bacterial biofilm formation by gallium nitrate is achieved through upregulation of *pykF* expression.

**1.3.2 *In vivo* study of *pykF's* role in biofilm formation in rat bone infection models.** To assess the effect of *pykF* knockout in *E. coli* on infection outcomes, *pykF*-deficient strains were introduced into the tibial marrow cavity and subcutaneous tissue of SD rats to establish a *KO-pykF-E. coli* bone infection model. The animals were followed for 12 weeks with continuous monitoring of body weight and temperature (modeling procedure shown in S3 Fig). Following

euthanasia, histological evaluation with H&E staining was performed to examine tibial bone pathology, peri-implant soft tissue alterations, and capsule thickness surrounding the implanted material. Body weight initially declined in both groups but gradually increased thereafter, without significant intergroup differences. Temperature measurements similarly showed no statistically significant variation (S3D Fig). Histological analysis revealed that tissues from the KO-*pykF*-*E. coli* group displayed more extensive structural alterations compared with those from the WT-*E. coli* group (S3E Fig). Examination of tibial sections further demonstrated aggravated tissue degeneration and more extensive biofilm accumulation at the margins in the KO-*pykF*-*E. coli* group relative to the WT-*E. coli* group (Fig 6A). Examination of perilesional soft tissue indicated marked structural disorganization, extensive necrosis, and dense infiltration of inflammatory cells in the *KO-pykF-E. coli* group compared with the WT-*E. coli* group (Fig 6B). Capsule thickness surrounding the implanted material was markedly greater in the KO-*pykF*-*E. coli* group than in the WT-*E. coli* group (Fig 6C). To assess whether *pykF* deletion influenced host inflammatory responses, serum IL-6, IL-1β, and TNF-α concentrations were quantified by ELISA, revealing substantially elevated levels in the KO-*pykF*-*E. coli* group relative to

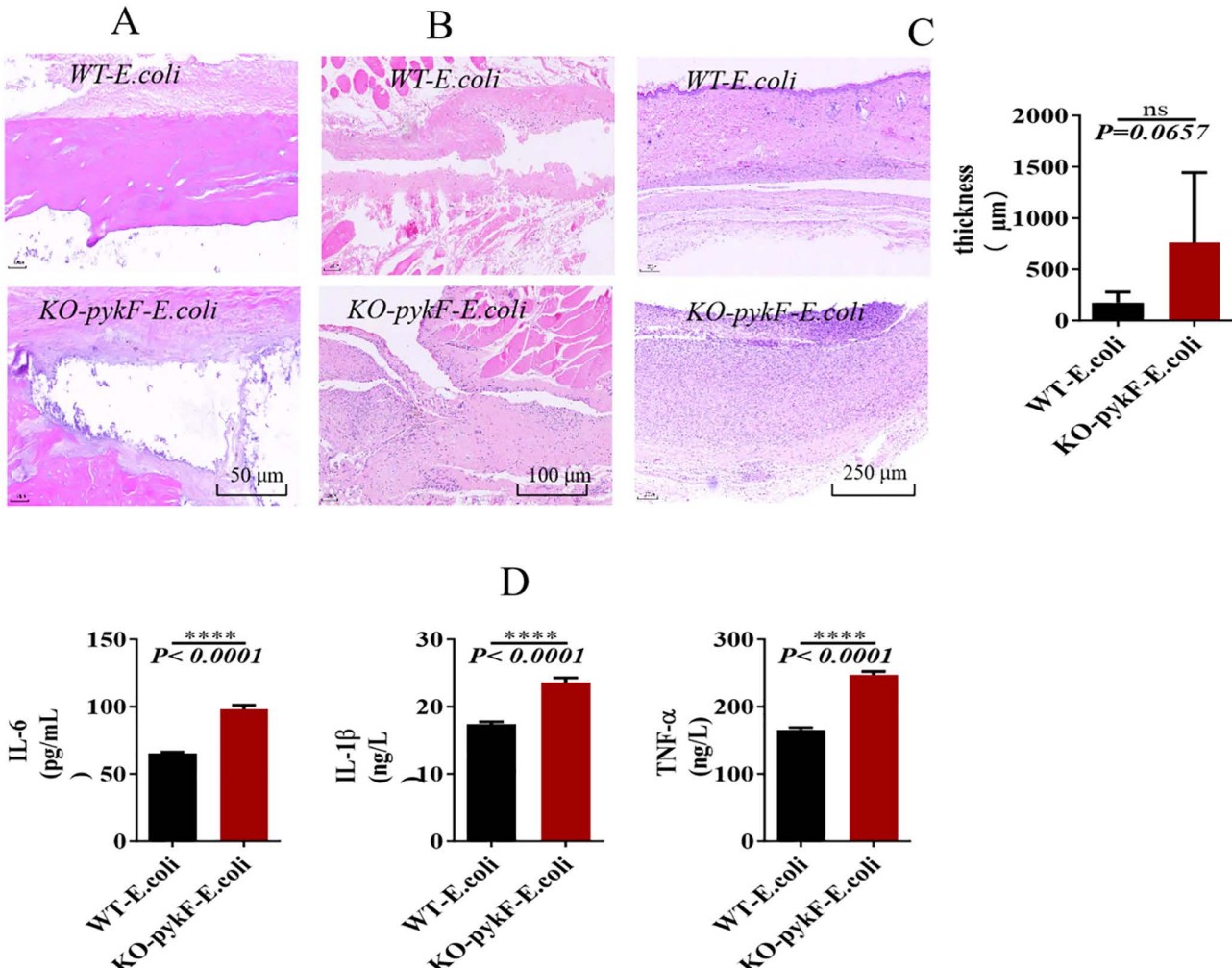

**Fig 6. Effect of *pykF* knockout in *E. coli* on rat infection.** A: hematoxylin and eosin (H&E) staining of tibial bone pathology (200× magnification). B: H&E staining of soft tissue alterations at the injection site (100× magnification). C: H&E staining for capsule thickness around the implanted material (40× magnification). D: ELISA quantification of serum interleukin (IL)-6, IL-1β, and tumor necrosis factor α.

the WT-*E. coli* group (P<0.0001; Fig 6D). Overall, *pykF* knockout in *E. coli* resulted in accelerated bacterial proliferation, increased biofilm formation, pronounced inflammatory cell infiltration, upregulation of proinflammatory cytokines, extensive cellular necrosis, and aggravated tissue infection in rats.

## 2. Conclusion

Rapid progress in modern medical technology has markedly increased the frequency of internal fixation procedures; however, bone implant infections remain a major obstacle to therapeutic success. Such infections prolong hospitalization, raise medical expenses, and severely diminish patient quality of life, thereby imposing heavy burdens on both families and healthcare systems. The challenge largely arises from biofilm-producing pathogens such as *Staphylococcus aureus*, *Escherichia coli*, and other Gram-negative bacteria [15,16]. Biofilm formation induces bacterial phenotypic adaptation, strengthens resistance to multiple antibiotics, and enhances protection against host immunity [17]. This multifactorial resistance frequently renders conventional therapeutic regimens ineffective, emphasizing the urgency of developing approaches that suppress bacterial biofilm formation for the prevention and treatment of implant-associated bone infections.

However, compared with conventional antibacterial metals such as silver, copper, and zinc, gallium exhibits greater stability and broader antimicrobial activity, making it a promising candidate for novel biomaterial applications. Gallium nitrate replaces iron in essential iron-dependent metabolic processes—including electron transport, DNA replication, and oxidative stress defense [5,6]—thereby disrupting bacterial physiology and limiting biofilm development. Under iron-restricted conditions, gallium nitrate further enhances activity against methicillin-resistant *Staphylococcus aureus* through complexation with gallium protoporphyrin [7]. In addition, combined administration with colistin suppresses the proliferation of multidrug-resistant *Klebsiella pneumoniae* [8]. Consistent with these effects, earlier work demonstrated that gallium nitrate-coated $TiO_2$ nanotubes substantially reduced mixed-species biofilms formed by *S. aureus* and *Escherichia coli* [12]. In this study, the optimal inhibitory concentration of gallium nitrate was applied to *E. coli, Pseudomonas aeruginosa*, and *S. aureus*. Crystal violet staining, XTT assay, confocal laser scanning microscopy (CLSM), and scanning electron microscopy (SEM) were employed to assess bacterial responses after treatment. Results demonstrated that gallium nitrate markedly reduced both biofilm formation and bacterial viability, consistent with previous reports.

Transcriptomic sequencing combined with bioinformatics analysis identified *pykF* and *pykA* as major targets associated with gallium nitrate–mediated biofilm inhibition. Validation at transcriptional and protein levels revealed more pronounced alterations in *pykF* compared with *pykA*. Previous studies have shown that pyruvate kinase encoded by *pykF* contributes to virulence regulation in *Vibrio vulnificus* through the Vvrr1–*pykF* axis, which influences adhesion, biofilm development, and pathogenicity [13]. Furthermore, Fang et al. reported that deacetylation of Lys413 on pyruvate kinase enhanced its catalytic efficiency, thereby elevating bacterial energy metabolism and increasing the susceptibility of resistant strains to antibiotics [14]. Protein acetylation thus acts as a dual regulator, repressing metabolic activity to adjust antibiotic resistance while enhancing motility. As a glycolytic enzyme central to bacterial metabolism, pyruvate kinase undergoes reversible deacetylation, which restricts growth by modulating metabolic flux.

Gallium citrate disrupts the metabolism of *Pseudomonas aeruginosa* by competing with iron uptake; however, its antibacterial efficacy is diminished due to rapid inactivation under physiological conditions [18]. Gallium maleate also demonstrates limited activity, largely attributable to its poor penetration of the *Staphylococcus aureus* cell membrane. In contrast, gallium nitrate exhibits superior stability in physiological environments because of the weak coordination of nitrate ions, which enables sustained release of $Ga^{3+}$. This property enhances its ability to disturb bacterial iron homeostasis and inhibit iron-dependent enzymatic processes [19]. Experimental evidence further indicates that gallium nitrate markedly suppresses multidrug-resistant *Acinetobacter baumannii* and polymyxin-resistant *Klebsiella pneumoniae* [19,20], reflecting a broader antibacterial spectrum and more consistent activity.

The *pykF* gene, encoding pyruvate kinase, occupies a central role in the glycolytic pathway of bacteria. In engineered *Escherichia coli* strains designed for 4-hydroxyisoleucine biosynthesis, deletion of *pykF* redirects phosphoenolpyruvate toward oxaloacetate, leading to a substantial increase in 4-hydroxyisoleucine yield and demonstrating its regulatory influence over intracellular carbon flux. Comparative studies in *Bacillus subtilis* have revealed distinct adaptive mechanisms of *pykF* regulation under environmental stress, differing from those observed in *Escherichia coli* [21]. Consequently, although *pykF* maintains a conserved metabolic function across bacterial taxa, its regulatory networks and adaptive responses are species-specific, reflecting variations in ecological niches, cellular organization, and metabolic strategies [22].

The *pykF* pathway appears to be influenced by off-target interactions and feedback regulation, with downstream metabolic enzymes such as PpsA and Eno representing likely participants in this control network. A recent study in *Escherichia coli* demonstrated that glycolysis generated minute-scale oscillations in metabolite levels, and deletion of *pykF* markedly intensified periodic oscillations of fluorescence resonance energy transfer (FRET) signals at the single-cell level, highlighting the role of *pykF* in stabilizing glycolytic dynamics and suggesting its involvement in metabolite-driven feedback regulation [23]. Nevertheless, interventions directed at *pykF* may inadvertently disrupt parallel metabolic routes. Comprehensive elucidation of this pathway will require integration of transcriptomic and metabolomic datasets to define its regulatory structure and to evaluate the extent of potential off-target effects.

This study sought to determine whether pyruvate kinase functions as a key target in gallium nitrate–mediated suppression of bacterial biofilm formation and activity. Comparison between *pykF*-knockout and wild-type Escherichia coli strains treated with gallium nitrate demonstrated that deletion of *pykF* alone enhanced both biofilm formation and bacterial viability. Moreover, removal of *pykF* markedly diminished the inhibitory effect of gallium nitrate on biofilm development and bacterial activity. The role of *pykF* in biofilm regulation observed here aligns with previous reports. To the best of current knowledge, this work provides the first evidence that gallium nitrate suppresses biofilm formation through modulation of *pykF* expression.

Bacterial biofilm formation constitutes a major etiological factor in capsular contracture [18], and effective control of implant-associated infections substantially reduces its incidence [19]. Capsular contracture is characterized by the development of dense and constrictive fibrotic tissue surrounding implants, frequently leading to tissue distortion, chronic pain [20], and persistent clinical management challenges. In this study, two rat models were employed: one replicating capsular contracture and another representing bone infection. Infection with *pykF*-deficient *Escherichia coli* markedly increased both capsule thickness and rigidity in the capsular contracture model. In the bone infection model, *pykF*-deficient *E. coli* induced similar fibrotic capsule layers, which aggravated the progression of bone infection. Collectively, the data indicate that *pykF* constitutes a potential therapeutic target for preventing capsular contracture and attenuating bone infection.

In summary, gallium nitrate markedly suppresses bacterial biofilm formation while exerting strong antibacterial effects. This work provides the first evidence that pyruvate kinase functions as a central target mediating the inhibitory action of gallium nitrate on biofilm development. In addition, *pykF* contributes to the regulation of capsular contracture and bacterial bone infections. Concurrent targeting of *pykF* and gallium may therefore offer a more effective therapeutic approach for refractory biofilm-associated bacterial infections, while also establishing a novel target for managing persistent infections sustained by biofilm formation.

Several limitations of this study require careful consideration. First, the relatively short observation period restricts evaluation of the long-term stability, safety, and antibacterial efficacy of gallium nitrate coatings in clinical applications. Second, the investigation was confined to *Staphylococcus aureus*, *Escherichia coli*, and *Pseudomonas aeruginosa*, leaving unanswered whether gallium nitrate coatings exert comparable antibacterial effects against other biofilm-forming pathogens, including coagulase-negative staphylococci and streptococci. Third, the *in vivo* observation period was insufficient to determine long-term outcomes and biosafety, and no systematic assessments of cytotoxicity or histocompatibility were

conducted, emphasizing the necessity for targeted evaluations in subsequent studies. Fourth, the limited spectrum of bacterial models restricts generalizability across diverse pathogenic species. Fifth, the molecular mechanisms through which *pykF* regulates biofilm formation remain undefined. Future work will concentrate on delineating the signaling pathways by which *pykF* modulates biofilm development, aiming to identify strategies for reducing biofilm formation and mitigating implant-associated infections. Addressing these limitations is essential for advancing the potential application of gallium nitrate and *pykF* in the prevention and management of biofilm-related infections.

## 3. Experimental section

### 3.1 Determination of the antimicrobial properties of gallium nitrate and its effectiveness against bacterial biofilms

**3.1.1 Bacterial culture.** *E. coli*, *S. aureus*, and *Pseudomonas aeruginosa* were purchased from the American Type Culture Collection (ATCC). 5 mL of tryptic soy broth was added to a centrifuge tube containing *S. aureus* (ATCC 25923). Similarly, 5 mL each of Luria–Bertani (LB) broth was added to centrifuge tubes containing *E. coli* (ATCC 25922) and *P. aeruginosa* (ATCC 27853). Carefully avoiding air bubbles, the freeze-dried strains were dissolved into a suspension by pipetting up and down with a sterile pipette tip. The bacteria were streaked LB agar plates and incubated at 37°C until colonies appeared, followed by P1 generation cultivation.

The cultured bacterial suspension was diluted to $1 \times 10^8$ colony-forming units (CFU)/mL. Gallium nitrate (Shanghai Mailin Biochemical Technology Co., Ltd.) was added to the bacterial suspension at concentrations of $1 \times 10^3$–$1 \times 10^5$ CFU/mL. After incubating on a shaker at 37°C for 24 h, 5 μL of the mixture was spread onto plates and incubated overnight at 37°C. The minimum inhibitory concentration (MIC) of gallium nitrate was observed, photographed, and evaluated.

**3.1.2 Crystal violet staining to observe biofilm formation ability.** Each bacterial suspension was diluted to $1 \times 10^5$ CFU/mL using gallium nitrate solution and placed in the wells of a 24-well plate slide. Bacterial suspensions diluted with phosphate-buffered saline (PBS; G0002-2L; Sevellar) were used as the control group. After incubation at 37°C for 72 h, the culture medium was removed and the slide was washed 1–2 times with PBS and fixed with 1 mL of 4% paraformaldehyde for 30 min. After removing the paraformaldehyde, the slide was washed 1–2 times with PBS and incubated for 1 h at 37°C. The wells were then stained with 0.4% crystal violet for 20 min and washed three times with PBS. The slide was allowed to air dry naturally and was photographed.

**3.1.3 XTT assay to determine the dynamics of bacterial biofilm formation.** We diluted each bacterial suspension to $1 \times 10^5$ CFU/mL using gallium nitrate solution and added each sample to a 24-well plate slide. A bacterial suspension diluted with PBS was used as the control group. After incubation at 37°C for 24 h, we added 20 μL of XTT working solution to each well and incubated in a constant temperature incubator at 37°C for 3 h. The optical density (OD) was measured at 450 nm using a microplate reader. The survival rate of each group was calculated as follows: bacterial survival rate (%) = (OD of treated bacteria/ OD of control bacteria) × 100.

**3.1.4 Scanning electron microscopy (SEM) to observe biofilm ultrastructure.** Each bacterial suspension was diluted to $1 \times 10^5$ CFU/mL using gallium nitrate solution and added to 24-well plate slides. Bacterial suspensions diluted with PBS were used as the controls. After incubation at 37°C for 24 h, we collected the freshly cultured bacteria, rinsed the slide with 1 × PBS (pH 7.0), and added 1 mL of 2.5% glutaraldehyde solution for fixation at 4°C for 4 h or overnight. Next, we added 500 μL of 1% osmium tetroxide, mixed thoroughly by shaking, and fixed the samples for 1–2 h. The slides were then dehydrated through an ascending gradient of ethanol solutions (20%, 50%, 80%, and 100%) for 10 min each, followed by transitioning to pure acetone through multiple changes of 100% ethanol. Finally, 200 μL of pure acetone was added and the samples were kept at 4°C for 20–30 min before being allowed to air dry naturally. The slides were sealed and stored at 4°C for observation using SEM (Apreo; Thermo Fisher Scientific).

**3.1.5 Transmission electron microscopy (TEM) observation of the internal structure of bacterial cells.** We collected freshly cultured bacteria and adjusted the concentration to $1 \times 10^8$–$1 \times 10^9$ cells/mL with sterile water. Then, we mixed the bacterial suspension with an equal volume of 2% sodium phosphotungstate aqueous solution to prepare a

mixed bacterial suspension. Using a sterile capillary tube, we dropped the mixed bacterial suspension onto a copper grid membrane. After 3–5 min, we removed excess water with filter paper. The sample was allowed to dry and then examined under a low-power optical microscope to select copper grids with intact membranes and evenly distributed bacterial cells, after which it was observed under a TEM (Talos F200X S/TEM; Thermo Fisher Scientific).

**3.1.6 Observation of the thickness and integrity of bacterial biofilms using confocal laser scanning microscopy (CLSM).** We diluted the bacterial suspension with gallium nitrate solution to $1 \times 10^7$ CFU/mL and added it to 24-well plate slides for culture at 37°C for 72 h. We used Live/Dead *Bac*Light Bacterial Viability Kit (L13152; Thermo Fisher Scientific) for the determination of the effect of gallium nitrate treatment on the three groups of bacteria. The kit includes two fluorescent staining solutions: SYTO9 (Solution A) and propidium iodide (PI; Solution B). SYTO9 emits green fluorescence (excitation wavelength, 488 nm; emission wavelength, 519 nm) in live bacteria, whereas PI emits red fluorescence (excitation wavelength, 559 nm; emission wavelength, 567 nm) in dead bacteria. Briefly, after washing with sterile PBS, 400 µL of the fluorescent dye mixture was added to each well and allowed to stain for 10 min at room temperature. The slides were then removed and gently rinsed in 2.5-mL physiological saline to remove excess dye. After sealing with Solution C, coverslips were placed on the slides and the bacteria were observed under a CLSM600 confocal laser scanning microscope (SOPTOP)1 with an argon ion laser as the light source. Observation parameters included measuring bacterial biofilm thickness, generating three-dimensional reconstructed images, and calculating the percentage of live bacteria on the biofilm surface at various time points based on the area of green (live) and red (dead) fluorescence. All observations were conducted in three fields per sample, with layer scanning performed to record depths. All procedures were performed under light-shielded conditions.

## 3.2 Transcriptome sequencing for potential targets of gallium nitrate inhibition in bacteria

**3.2.1 Experimental procedure.** The total RNA of the samples was isolated and purified using TRIzol reagent (15596018; Thermo Fisher Scientific) according to the manufacturer's instructions. The quantity and purity of total RNA from two replicates each of the *E. coli* control and treatment groups were assessed using a NanoDrop ND-1000 spectrophotometer (Wilmington, DE, USA), and RNA integrity was evaluated using a 2100 Bioanalyzer System (Agilent, CA, USA). Samples with concentrations > 50 ng/µL, an RNA integrity number > 7.0, and total RNA > 1 µg were deemed suitable for downstream experiments. Polyadenylated mRNA was selectively captured using oligo(dT) magnetic beads (Dynabeads Oligo (dT)$_{25}$, 61005; Thermo Fisher Scientific) with two rounds of purification. The captured mRNA was then fragmented at 94°C for 5–7 min using the NEBNext Magnesium RNA Fragmentation Module (E6150S; New England Biolabs, USA). The fragmented RNA was reverse transcribed into cDNA using SuperScript II Reverse Transcriptase (1896649; Invitrogen, CA, USA). Subsequently, double-stranded DNA was synthesized using DNA polymerase I (*E. coli*) (m0209; New England Biolabs), with RNase H (m0297; New England Biolabs) used to convert the DNA–RNA hybrid double strands into double-stranded DNA. During this process, dUTP solution (R0133; Thermo Fisher Scientific) was incorporated to blunt the ends of the double-stranded DNA, followed by the addition of an A base to each end to enable connection with adapters bearing T bases. The fragment sizes in the library were selected and purified using magnetic beads. After digestion of the double strands with uracil DNA glycosylase (m0280; New England Biolabs), polymerase chain reaction (PCR) was performed as follows: 3 min of predenaturation at 95°C; eight cycles of denaturation at 98°C for 15 s, annealing at 60°C for 15 s, and extension at 72°C for 30; and final extension at 72°C for 5 min. The resultant library comprised fragment sizes of $300 \pm 50$ bp (strand-specific library). Finally, paired-end 150 sequencing was performed on an Illumina NovaSeq 6000 System by Hangzhou Lianchuan Biotechnology (Hangzhou, China).

**3.2.2 Reliability analysis.** Plane raw data format for fastq, use the fastp (https://github.com/OpenGene/fastp) software to offload raw data for quality control, including removal of joint, repeat sequence, and low quality parameters

as the default. Use HISAT2 (https://ccb.jhu.edu/software/hisat2) will be sequenced data comparing to the genome (GCF_000005845. 2 _asm584v2), file format for bam. Using StringTie software (https://ccb.jhu.edu/software/hisat2) on gene transcription or the assembly using FPKM quantitative (FPKM = total_exon_fragments/ mapped_reads(millions) × exon_length(kB)]), Using R package edgeR (https://bioconductor.org/packages/release/bioc/html/edgeR.html) to analyze the difference between sample genes, Differential genes were defined as |log2FC| >= 0.5 and p < 0.05.

**3.2.3 Reverse transcription PCR (RT-PCR).** Total RNA was extracted separately from each group and subjected to RT-PCR. Subsequently, quantitative PCR (qPCR) was performed using 2 × Universal Blue SYBR Green qPCR Master Mix (BZ2106008; Servicebio). The qPCR process was as follows: 1 min of initial denaturation at 95°C, followed by 40 cycles of denaturation at 95°C for 20 s, annealing at 55°C for 20 s, and extension at 72°C for 30 s. Table 1 lists the primer sequences.

**3.2.4 Western blot (WB).** After thawing the samples on ice for 10 min, they were centrifuged at 14,000 × g for 15 min at 4°C. The protein concentrations were determined using a BCA Protein Assay Kit (P0012; Beyotime). Subsequently, 80 μL of protein was mixed with 20 μL of 5 × protein loading buffer, boiled in a water bath for 5 min, and then subjected to sodium dodecyl sulfate-polyacrylamide gel electrophoresis. Next, the gel was transferred onto a polyvinylidene fluoride membrane. The membrane was cut into strips for each lane and incubated in blocking solution (5% skim milk) on a shaker at room temperature for 40 min. The primary antibody against *pykF* (CSB-EP364920ENV; Cusabio) was added and incubated overnight at 4°C, followed by incubation with a secondary antibody for 40 min, development, and photographic documentation. ImageJ v1.8.0.345 software was used for quantitative analysis of the WB bands, where the relative protein quantification was determined by dividing the grayscale value (max–mean) of the target protein by that of the reference protein.

## 3.3 Effect of gallium nitrate on *pykF*-knockout strains

**3.3.1 *In vitro* tests of gallium nitrate's effects on *pykF mRNA* and biofilm.** Wild-type and *pykF*-knockout strains of *E. coli* were treated with gallium nitrate. Then, the impact of gallium nitrate and *pykF*-knockout intervention on *pykF* mRNA expression levels was assessed using qPCR, and the impact on pyruvate kinase levels was assessed using WB. Subsequently, following methods outlined in sections 3.1.4–3.1.6, CLSM was used to observe the influence of gallium nitrate treatment and *pykF* knockout on the thickness and capability of biofilm formation. SEM and TEM were used to examine changes in the biofilm ultrastructure.

**3.3.2 Rat intraosseous infection models to study *pykF's* effect on biofilm.**

Model 1 (bone infection model):

*pykF*-knockout strains of *E. coli* were inoculated into the marrow cavity of the tibia of 8-week-old male Sprague–Dawley (SD) rats (Production License Number: SCXK (Beijing) 2019−0010, Use License Number: SYXK (Yunnan) K2020-0006; Beijing Huafukang Biotechnology Co., Ltd.), followed by implantation of titanium needles and wound closure using bone wax and layered sutures. The control group was inoculated with normal *E. coli* at the same site. All rats were allowed access to food and water ad libitum upon recovery from anesthesia. Body weight was monitored weekly. At 12 weeks

**Table 1. PCR primer sequences.**

| Gene | Forward (5′–3′) | Reverse (5′–3′) |
| --- | --- | --- |
| *16sRNA* | CTGGAACTGAGACACGGTCC | GGTGCTTCTTCTGCGGGTAA |
| *rpoA* | GCACCAAAGAAGGCGTTCAG | ATATCGGCTGCAGTCACAGG |
| *PykF* | CTAAAATGCTGGACGCTGGC | GCGGCGGTTTTACCAGTTTT |
| *pykA* | CAAAACTGGGGCGTCATGTG | GTCGCCTTCACCTTTACCCA |

postsurgery, the rats were euthanized and the subfascial tissue surrounding the tibia and tissue around the needle insertion site were collected. Pathological changes in the tibia were examined using hematoxylin and eosin (H&E) staining, and those in the soft tissue at the needle insertion site were also evaluated. Finally, enzyme-linked immunosorbent assay (ELISA) kits were used to measure the serum levels of interleukin (IL)-6 (MM-0047R2), IL-1β (MM-0190R2), and tumor necrosis factor α (TNF-α) (MM-0180R2).

Model 2 (capsular contraction model):

SD rats (male, 8 weeks old) were purchased from Beijing Huafukang Biotechnology Co., Ltd. (Production License Number: SCXK (Beijing) 2019−0010, Use License Number: SYXK (Yunnan) K2020-0006). Two points were marked on each side of the midline of the back, and four longitudinal incisions (approximately 1-cm long) were made. Subfascial tissue was bluntly dissected at each surgical incision. Polydimethylsiloxane was placed subcutaneously in the incisions, which were then sutured closed. Immediately after placement, 0.1 mL of *pykF*-knockout *E. coli* suspension was injected into the skin adjacent to the surgical incisions where the material was implanted. The control group was injected with normal *E. coli* at the same site. Rat body weight was monitored weekly. At 12 weeks postsurgery, the rats were euthanized and tissue surrounding the implant was collected. Pathological changes in the tissue surrounding the implant were examined using H&E staining. Finally, ELISA kits were used to measure the serum levels of IL-6, IL-1β, and TNF-α.

### 3.4  Statistical analysis

Statistical analysis was conducted using GraphPad Prism (10.1.2) software. Experimental data were presented as the mean ± standard error of the mean. Between-group comparisons of continuous data were performed using the Student's *t*-test. Comparisons among multiple groups were analyzed using one-way analysis of variance. $p < 0.05$ was considered statistically significant, $p < 0.01$ was considered highly significant, and $p < 0.001$ was considered extremely significant.

### Supporting information

**S1 Fig. Screening of the optimal GA intervention concentration.** Concentrations tested included 4096 μg/ml, 2048 μg/ml, 1024 μg/ml, 512 μg/ml, 256 μg/ml, 128 μg/ml, 64 μg/ml, 32 μg/ml, and 8 μg/ml. The bacterial suspension was diluted with gallium nitrate at 1:1000 to yield a final concentration of $1 \times 10^5$ CFU/ml. After incubation at 37°C for 24 hours, cultures were photographed and examined.
(DOCX)

**S2 Fig. (A) Strategy for generating the *pykF* knockout (KO) strain in *E. coli*.** (B) Combined effects of gallium nitrate (GA) exposure and *pykF* deletion on *pykF* expression. (a) RT-PCR analysis of *pykF* mRNA levels in *E. coli* with rpoA as internal reference. The relative fold changes were as follows: KO-*pykF* vs. *KO-NC*, 0.552; KO-*pykF*+GA vs. *KO-NC*+GA, 0.279; KO-*pykF*+GA vs. KO-*pykF*, 1.094. (b) Western blot analysis of *pykF* protein expression with GAPDH as internal control. Relative fold changes were: *KO-pykF* vs. *KO-NC*, 0.250; *KO-pykF*+GA vs. *KO-NC*+GA, 0.474; *KO-pykF*+GA vs. *KO-pykF*, 2.570. Significance levels: *P < 0.05, **P < 0.01, ***P < 0.001, ****P < 0.0001.
(DOCX)

**S3 Fig. *In vivo* evaluation of *pykF* in biofilm formation using a rat bone infection model.** (A) Experimental timeline: bacteria cultured at week 0; animal modeling at week 1; body weight and temperature monitored weekly through week 10; animals maintained until week 12, when euthanasia and retrieval of polydimethylsiloxane implants were performed. (B) Culture outcomes of WT-*E. coli* and KO-*pykF-E. coli* groups. (C-a) Tibial modeling procedure. (C-b) Subcutaneous dorsal modeling procedure. (D) Longitudinal changes in body weight (left) and temperature (right) of rats infected with WT-*E. coli*

or *KO-pykF-E. coli* from weeks 0–10. (E) Polydimethylsiloxane implants collected from dorsal subcutaneous sites of rats in WT-*E. coli* and *KO-pykF-E. coli* groups.
(DOCX)

## Author contributions

**Conceptualization:** xiaofeng zhang.

**Data curation:** junjie dong.

**Formal analysis:** Bing Wang.

**Funding acquisition:** Lingqiang Chen.

**Investigation:** Zhiqiang Gong.

**Project administration:** Jin Yang.

**Resources:** Guizhao Shu.

**Validation:** Qi Ning.

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
