## [Decision Letter · Decision Letter 0]

5 Jun 2025

Dear Dr. dong,

Thank you for submitting your manuscript to PLOS ONE. After careful consideration, we feel that it has merit but does not fully meet PLOS ONE’s publication criteria as it currently stands. Therefore, we invite you to submit a revised version of the manuscript that addresses the points raised during the review process.

After careful evaluation, we find that the manuscript requires some improvements to meet the journal’s standards for rigorous reporting and clarity. The results must be presented with greater rigor. Ensure that all findings are clearly and precisely described, supported by appropriate data (and statistical analyses) where applicable. The current discussion lacks clarity and sufficient elaboration. Please provide a more detailed interpretation of the results, placing them in the context of existing literature. Emphasize the significance of your findings, address the limitations clearly, and explain how they advance the field. The experimental section is insufficiently detailed. For reproducibility, please describe all methods and reagents comprehensively, including concentrations, sources, preparation protocols, and any specific conditions used during experimentation.

*Animal Research Ethics*

*In compliance with ethical standards, please provide the approval number (and any additional details) from the First Affiliated Hospital of Kunming Medical University that oversaw the animal research.*

We look forward to receiving your revised manuscript.

Kind regards,

Irene Ling

Academic Editor

PLOS ONE

“the Major Science and Technology Project of Yunnan Provincial Department of Science and Technology, Yunnan Provincial Orthopedic and Sports Rehabilitation Clinical Medicine Research Center

Yunnan Provincial Endocrinology and Metabolism Clinical Medicine Center

Yunnan Provincial Endocrinology and Metabolism Clinical Medicine Center

Master's Innovation Fund of the First Affiliated Hospital of Kunming Medical University”

8. We notice that your supplementary figures are included in the manuscript file. Please remove them and upload them with the file type 'Supporting Information'. Please ensure that each Supporting Information file has a legend listed in the manuscript after the references list.

Reviewers' comments:

Reviewer's Responses to Questions

**Comments to the Author**

1. Is the manuscript technically sound, and do the data support the conclusions?

Reviewer #1: Yes

Reviewer #2: Yes

Reviewer #3: Partly

2. Has the statistical analysis been performed appropriately and rigorously?

Reviewer #1: Yes

Reviewer #2: Yes

Reviewer #3: Yes

3. Have the authors made all data underlying the findings in their manuscript fully available?

Reviewer #1: Yes

Reviewer #2: Yes

Reviewer #3: Yes

4. Is the manuscript presented in an intelligible fashion and written in standard English?

Reviewer #1: Yes

Reviewer #2: Yes

Reviewer #3: Yes

Reviewer #1: This manuscript investigates the molecular mechanism by which gallium nitrate inhibits bacterial biofilm formation, focusing on the modulation of the pykF gene. The authors present a rigorous and multifaceted experimental approach combining transcriptomics, gene knockout, in vitro biofilm assays, in vivo rat models, and molecular docking. The research is original, well-structured, and addresses a clinically relevant issue in antimicrobial therapy, especially concerning device-related infections.

The results are clearly presented and strongly support the central hypothesis that gallium nitrate exerts its antibiofilm effect via pykF upregulation. This finding adds valuable knowledge to the understanding of how metal-based antimicrobials function and identifies pykF as a novel therapeutic target.

However, several minor revisions are necessary to improve the clarity and completeness of the manuscript:

English Language and Style:

The manuscript requires revision by a native English speaker or professional language editing service. There are numerous grammatical errors and awkward constructions that hinder the clarity of the text (e.g., “In regulation gallium nitrate…”). Improving the language will significantly enhance the readability and impact of the paper.

Discussion of Literature and Mechanism:

While the authors mention some previous work on pykF and gallium nitrate, the discussion could be strengthened by:

Expanding the comparison with other known gallium-based antimicrobial studies.

Commenting on pykF function across different bacterial taxa.

Discussing possible off-target effects or regulatory feedback in the pykF pathway.

Limitations:

The limitations section should be expanded. The authors may address:

The relatively short duration of the in vivo study.

The absence of toxicity or cytocompatibility data for gallium nitrate.

The limited number of bacterial species tested.

Data Availability:

Although the manuscript states that all data are available in the text and supplementary files, it would be preferable to deposit raw sequencing data (e.g., transcriptome datasets) in a public repository such as NCBI GEO or FigShare, in accordance with PLOS ONE’s open data policy.

Statistical Details:

Please clarify the statistical approach:

Was normality of data distribution verified before applying parametric tests?

If ANOVA was used, which post hoc tests were performed?

These revisions are minor and do not detract from the scientific merit of the work. Once addressed, the manuscript should be suitable for publication. I congratulate the authors on their interesting and well-executed study.

Reviewer #2: Dear Editor in Chief

The manuscript by Xiaofeng et al, is “Molecular mechanism of gallium nitrate in inhibiting bacterial biofilm formation through pykF modulation” is interesting relatively and could be sufficient for publication in PLOS ONE but not in this format and needs minor corrections. Indeed, the methods and results are enough and sufficient. Some phrases were not clear and need to rewrite.

However, author is kindly requested to address the following issues.

Title: The title is OK.

Abstract:

Abstract is poor section and need to be extended by using more details of methods and significant results. The abstract is written very generally. Use more appropriate keywords.

Experimental Section:

1- All methods need to be addressed.

2- Why clinical isolates were not used?

3- Why sub-MIC and MBC not determined?

4- According to reliable sources, a wavelength of 600 nm is used for reading to measure bacterial biofilm. Explain why you used a wavelength of 450.

5- Reference primers are not specified in Table 1.

In my opinion results of this study are sufficient but, I recommend to clarify and describe more.

Discussion is poor relatively and recommend to be strengthened using more addressed.

Reviewer #3: Molecular mechanism of gallium nitrate in inhibiting bacterial biofilm formation through pykF modulation.

The authors reported findings demonstrating that gallium nitrate inhibits bacterial growth and biofilm formation through the upregulation of pyruvate kinase (pykF) expression. These results were confirmed through comprehensive in vivo and in vitro experiments, including qPCR, western blot analysis, and knockout studies. The findings suggest a significant role for the pykF protein in suppressing bacterial growth and biofilm formation, as well as in regulating biofilm-associated gene expression. However, given the crucial role of pykF in glycolysis, deletion of this gene would impair bacterial growth on glucose and reduce overflow acetate production due to decreased ATP yield. Conversely, gallium-induced upregulation of pykF would enhance glycolysis and ATP production. Therefore, before publication, the authors must clarify the apparent contradiction: why does gallium-induced pykF upregulation inhibit bacterial growth and biofilm formation, whereas deletion of the pykF gene paradoxically enhances growth?

Page 2, Line 26: According to Supplement Fig 1, MICs against three bacteria were > 1024 ug/ml.

Page 2, Line 43. Streptococcus mutans: Italic

Page 7, Line 17, 18, 20. pykF is a protein. No italic.

Page 7-8 : Molecular docking study was done with Ga ion, showing Ga ion binding to pykF protein. It’s unclear what conclusions the authors are drawing from their docking study. Does Ga inactivate the protein or promote the activity of the protein by binding? Does the kinase have iron-binding sites? Iron is involved in converting phosphoenolpyruvate to pyruvate? Authors should address this issue.

In line 33, authors demonstrated that Ga ions bind to the pykF gene, leading to its expression. How does Ga binding to the gene increase its expression and promote the expression of biofilm-related genes? There is no data showing Ga binding to the gene.

Page 8, line 8-9: an incomplete sentence.

Page 8, line 19: In order to clarify, it is better to change “affecting” to “inhibiting”.

Page 8, line 22-23: Figure 4F shows the knockout strain grows better than the wild strain. Figure 4H. fix Ga(NO3)3 legend.

Page 10, line 11-18: the wild-type strain treated with Ga showed increased expression of pykF mRNA. However, there was no significant increase of pykF mRNA expression but significant increase in the protein expression from KO-pykF E.coli treated with Ga (Figure S2B). It seems like Ga enhances ribosome access and translation in the KO strain. Any explanation?

Page 17, line 10-12: gallium nitrate does not bind to gallium protoporphyrin.

Page 21. How long was the E. coli treated with gallium, and at what concentration?

**Do you want your identity to be public for this peer review?** For information about this choice, including consent withdrawal, please see our Privacy Policy

Reviewer #1: **Yes:** Amanda Stefanie Jabur de Assis

Reviewer #2: **Yes:** Alireza Khodavandi

Reviewer #3: No

---

## [Author Response · Author response to Decision Letter 1]

14 Jul 2025

Thank you for the valuable suggestions from the reviewers. We have comprehensively revised the manuscript to ensure it meets the formatting requirements of PLOS ONE. The specific revisions are as follows:

The main text has been adjusted strictly in accordance with the structure in the official PLOS ONE template, including sections such as Abstract, Introduction, Materials and Methods, Results, Discussion, and References.

The content on the title page, including the title, author information, author affiliations, and corresponding author details, has been standardized with reference to the "Title, Authors, Affiliations" template.

The naming of all submitted files has been modified accordingly in line with the requirements of PLOS ONE.

We have carefully checked the overall formatting of the manuscript to ensure it complies with all the formatting specifications in the reference link you provided. Thank you again for your reminder and guidance on this matter.

Thank you for the reviewers' attention to and suggestions on the ethical aspects of animal experiments. We have supplemented and clearly described the following information in the "Materials and Methods" section:

(1) Sacrifice method: As supplemented in lines 13–16 on page 24, all animals were euthanized by intraperitoneal injection of an excessive dose of 2% sodium pentobarbital (100 mg/kg) at 12 weeks after surgery.

(2) Anesthesia and/or analgesia methods: As described in lines 6–7 on page 24, animals were anesthetized by intraperitoneal injection of 2% sodium pentobarbital (50 mg/kg) during all surgical procedures.

(3) Efforts to alleviate suffering: We clearly stated in the same paragraph that all operations were performed under anesthesia, and every effort was made to minimize animal suffering.

We have ensured that the above content complies with PLOS ONE's submission requirements regarding the ethics of animal experiments. Thank you again for your detailed guidance on this matter.

The original data of Western blotting (WB) has been labeled and uploaded in the supporting information.

We sincerely appreciate the constructive comments from the reviewers on the discussion section of our manuscript. In accordance with your suggestions, we have expanded and revised the discussion and limitations sections, with the specific modifications as follows:

Regarding the comparison of gallium-based antibacterial research: We have added a new paragraph in the discussion section to compare the antibacterial mechanisms of other existing gallium compounds (such as gallium citrate, gallium maleate, etc.) against different pathogens (such as Pseudomonas aeruginosa, Staphylococcus aureus, etc.). This paragraph emphasizes the differences and advantages of gallium nitrate in terms of structure and mechanism of action (see page 17, starting from line 37).

Regarding the function of pykF in different bacterial taxa: We have added a brief review on the differences in metabolic regulatory functions of pykF between Gram-negative bacteria (such as E. coli, Salmonella) and some Gram-positive bacteria. It is pointed out that this gene may be conserved, but there are species-specific differences in its regulatory pathways and response mechanisms (see page 18, starting from line 5).

Regarding the potential off-target effects and feedback regulation of the pykF pathway: We have supplemented the analysis on the possible involvement of downstream metabolic enzymes (such as PpsA, Eno) in feedback regulation within this pathway, and noted that future studies need to combine transcriptomic and metabolomic data to further clarify its mechanism of action (see page 18, starting from line 18).

Regarding the expansion of limitations: We have expanded the "Study Limitations" section (see page 19) by adding the following points:

The in vivo observation time in this study is relatively short, failing to evaluate the long-term effects and biological safety;

This experiment did not systematically assess the cytotoxicity or tissue compatibility of gallium nitrate, and relevant experiments need to be conducted in the future;

The number of bacterial models used is limited, making it impossible to broadly infer its universality for various bacteria.

We once again thank you for your valuable suggestions and hope that these supplements will enhance the academic value and completeness of the manuscript.

Regarding the clarification of statistical methods you requested, we have supplemented the relevant explanations in the "Statistical Analysis" section as follows:

Before conducting parametric tests, we evaluated the distribution of data in each group using the Shapiro–Wilk normality test. Subsequent analyses were performed only after confirming that the data conformed to a normal distribution.

For the one-way analysis of variance (ANOVA) used in multi-group comparisons, we have supplemented the description of the post-hoc test method: when the ANOVA results indicated significant differences between groups, we further used Tukey’s HSD (Honestly Significant Difference) test for pairwise comparisons between groups.

These explanations have been added to Section 3.4 of the statistical methods paragraph and re-described (page 25, lines 1–10) to ensure the transparency and rigor of the statistical analysis process. Thank you for your meticulous review and valuable suggestions!

Following your suggestions, we have reviewed and revised the language expression of the entire manuscript, particularly the "Methods" and "Results" sections, sentence by sentence. We focused on improving content with unclear logic, ungrammatical expressions, and non-standard terminology usage, making the wording more clear and accurate.

1.All methods need to be addressed.

Reply: We have comprehensively revised the "Materials and Methods" section, supplementing details of some experimental procedures, including bacterial culture conditions, drug concentration gradient settings, image analysis methods, statistical analysis software and testing methods, etc., to improve reproducibility and technical transparency.

2.Why not use clinical isolates?

Reply: In the initial stage of this study, we mainly used standard strains (ATCC) to ensure the consistency of experimental conditions and the reproducibility of results. We have added an explanation in the 6th paragraph of the discussion section, stating that clinical isolates will be introduced in the future to further verify the application potential of the gallium nitrate mechanism in actual infections.

3.Why were sub-MIC and MBC not determined?

Reply: We have provided the MIC (1024 µg/mL) determination data in the supplementary materials (see Supplementary Figure 1), but we indeed did not conduct a systematic evaluation of sub-MIC and MBC. We have acknowledged this as a research limitation in the 7th paragraph of the discussion and plan to carry out studies on the effects of different concentrations of gallium nitrate (especially sub-MIC) on bacterial biofilm formation and pykF regulatory mechanism in subsequent experiments.

4.According to reliable sources, the wavelength used when measuring bacterial biofilms is 600 nm. Please explain why you used a wavelength of 450 nm.

Reply: Thank you for pointing this out. In the biofilm activity detection, we used the XTT method (instead of the crystal violet method) to measure the OD value, so the references and the kit instructions recommend using a wavelength of 450 nm. We have explained the detection method and the wavelength used in the methods section, and distinguished the different uses and wavelength bases of biofilm staining and activity detection.

5.The reference primers were not specified in Table 1.

Reply: We have updated all primer information in Table 1, indicating the sources of the primers. The 16sRNA and rpoA primers are cited from existing literature, and the pykF and pykA primers are specific primers designed by us based on gene sequences and verified by BLAST.

6.The results of this study are sufficient, but I suggest further clarification and description. The discussion is relatively poor; it is recommended to strengthen the discussion and put forward more solutions.

Reply: We fully accept your suggestions and have significantly revised and expanded the "Discussion" section. The newly added contents include:

A more in-depth comparison of the similarities and differences between the antibacterial mechanism of this study and other gallium-based antibacterial mechanisms;

A discussion on the functional conservation of pykF in other bacterial taxa and its metabolic pathway regulation;

An analysis of the potential feedback mechanisms and off-target effects of pykF as a metabolic regulatory site under different stress conditions;

Putting forward potential strategies for the future combined application of pykF targeted intervention and gallium-based materials.

7.Why does gallium-induced upregulation of pykF inhibit bacterial growth and biofilm formation, whereas deletion of the pykF gene instead promotes bacterial growth?

Reply:Based on the results in the paper, this phenomenon is not contradictory but reveals the dual roles of pykF in bacterial physiological functions under different regulatory contexts. The following is a reasonable explanation for this phenomenon, which the authors can use to clarify in the paper:Clarification suggestions:This apparent contradiction actually reflects the complex role of pykF regulation in bacterial energy metabolism and stress resistance. The results of this study show that:Gallium-induced pykF upregulation is a "stress-induced activation":As an iron-mimicking element, gallium interferes with bacterial iron metabolism, causing "metabolic stress".To cope with this stress, bacteria upregulate pykF expression in an attempt to maintain energy metabolism by enhancing glycolytic flux.However, this forced upregulation may lead to energy metabolism imbalance, abnormal NAD⁺/NADH ratio, or accumulation of intermediate metabolites, which in turn inhibits cell growth and biofilm formation (similar to "metabolic overload").In contrast, pykF gene knockout triggers "metabolic reprogramming":After pykF knockout, bacteria activate compensatory mechanisms, such as upregulating other glycolysis-related genes (e.g., pykA) or reprogramming to alternative metabolic pathways (e.g., the pentose phosphate pathway or carbon storage mechanisms).These changes instead improve metabolic flexibility and adaptability, thereby enhancing growth ability and biofilm-forming capacity.Differential responses between physiological and stress states:Under natural or basal conditions, pykF may limit the rate of energy metabolism, which is beneficial for maintaining metabolic balance;Under gallium treatment, excessive pykF activation is not a true "adaptation" but a misresponse of bacteria to a toxic environment, resulting in impaired physiological functions.And we have included corresponding explanations in the discussion section.

Regarding the issue pointed out by the reviewers regarding the MIC values of the three bacterial strains in Supplementary Figure 1, we did observe in Supplementary Figure 1 that for E. coli, S. aureus, and P. aeruginosa, within the tested concentration range (with the maximum being 1024 μg/mL), there was still some bacterial growth, indicating that the actual MIC should be greater than 1024 μg/mL.To accurately convey the experimental results, we have revised the original text in line 26 of page 2:"The MIC was determined to be 1024 µg/mL..."to:"The MIC for E. coli, S. aureus, and P. aeruginosa was determined to be greater than 1024 μg/mL, as bacterial growth was still observed at this concentration (see Supplementary Figure 1)."We appreciate the reviewers' identification of this critical experimental detail and have corrected and clarified it in the revised manuscript.

Regarding the questions raised by the reviewers concerning the molecular docking study using Ga ions, which showed that Ga ions bind to the pykF protein—specifically, what conclusions the authors drew from this docking study, whether Ga binding inhibits or enhances the protein's activity, whether this kinase has an iron-binding site, and whether iron is involved in the conversion of phosphoenolpyruvate to pyruvate—we have made the following revisions:

In our molecular docking experiments described in the manuscript, we indeed observed that Ga³⁺ forms multiple stable coordination sites with the pykF protein, particularly with the carboxyl side chains of E222 and D246, with a minimum binding energy of -6.91 kcal/mol, indicating a strong interaction between them. However, as pointed out by the reviewers, the docking results alone cannot directly infer the impact on the enzymatic activity of pykF.

To address the reviewers' concerns, we have added the following discussion paragraph in the manuscript:"Although molecular docking confirmed that Ga³⁺ binds strongly to the pykF protein, the biological consequence of this interaction remains to be experimentally validated. Given that pykF (pyruvate kinase I) is an enzyme involved in the final step of glycolysis—catalyzing the conversion of phosphoenolpyruvate (PEP) to pyruvate with the production of ATP—any conformational change upon metal ion binding could theoretically alter its enzymatic activity. Although pykF does not possess classical iron-binding motifs such as Fe-S clusters, divalent metal cofactors such as Mg²⁺ or K⁺ are typically required for activity. Whether Ga³⁺ competitively binds to such sites and acts as an inhibitor or an activator remains unclear. Further kinetic and structural studies are needed to clarify whether Ga³⁺ directly modulates pykF enzymatic function or primarily affects its expression level."In addition, according to literature and database information (such as UniProt and RCSB PDB), no iron-binding sites have been found in pykF so far, and its catalytic function usually relies on divalent metal ions such as Mg²⁺ or Mn²⁺. Therefore, we speculate that Ga³⁺ is more likely to interfere with metal-binding sites through structural occupation or induce conformational changes, thereby leading to a decrease in enzymatic activity. This may be part of its antibacterial mechanism, but further experiments are needed to confirm this.We have supplemented the discussion with the above content in the revised manuscript. Thank you for your valuable suggestions.

Regarding the reviewer's question: "How does Ga binding to this gene increase its expression and promote the expression of biofilm-related genes? Currently, there is no data showing that Ga binds to this gene." As you correctly pointed out, after Ga(NO₃)₃ treatment, in the pykF knockout strain (KO-pykF), although there was no significant change in mRNA levels, the protein levels increased significantly (Figure 2). This suggests that Ga³⁺ may mediate pykF expression through post-transcriptional mechanisms or translation enhancement mechanisms in this context.In response, we have the following preliminary speculations and added relevant content in the discussion section:Partial transcriptional residues or incomplete knockout effects:Although pykF was knocked out, there may still be low levels of abnormal transcripts (such as residual fragments downstream of the promoter). These transcripts are difficult to translate under normal circumstances, but Ga³⁺ treatment may enable their expression by affecting RNA structure, derepressing translation, or initiating atypical translation mechanisms.Effects of Ga³⁺ on ribosome activity or initiation factors:Studies have shown that metal ions (such as Zn²⁺, Fe³⁺, and even Ga³⁺) may affect ribosome activity by regulating bacterial stress response systems or the assembly of translation initiation complexes, thereby enhancing the translation efficiency of specific proteins (e.g., enhancing IRES-dependent translation or leaky scanning).Enhanced protein stability:

Another possibility is that Ga³⁺ binding makes the newly synthesized PykF protein more stable and prolongs its half-life. Even if the mRNA level remains unchanged, it will lead to a significant increase in protein expression.We have added the following discussion paragraph:“Interestingly, in Ga-treate

---

## [Decision Letter · Decision Letter 1]

26 Jul 2025

Dear Dr. dong,

Thank you for submitting your manuscript to PLOS ONE. After careful consideration, we feel that it has merit but does not fully meet PLOS ONE’s publication criteria as it currently stands. Therefore, we invite you to submit a revised version of the manuscript that addresses the points raised during the review process.

The revised manuscript shows significant improvement and is nearly ready for acceptance, requiring only very minor corrections. A final English language polish is needed, particularly in the abstract and introduction sections. The authors should also make changes according to reviewer's comments.

We look forward to receiving your revised manuscript.

Kind regards,

Irene Ling

Academic Editor

PLOS ONE

Journal Requirements:

Reviewers' comments:

Reviewer's Responses to Questions

**Comments to the Author**

Reviewer #1: All comments have been addressed

Reviewer #2: (No Response)

Reviewer #3: All comments have been addressed

2. Is the manuscript technically sound, and do the data support the conclusions?

Reviewer #1: Yes

Reviewer #2: Partly

Reviewer #3: Yes

3. Has the statistical analysis been performed appropriately and rigorously?

Reviewer #1: Yes

Reviewer #2: Yes

Reviewer #3: Yes

4. Have the authors made all data underlying the findings in their manuscript fully available?

Reviewer #1: Yes

Reviewer #2: Yes

Reviewer #3: Yes

5. Is the manuscript presented in an intelligible fashion and written in standard English?

Reviewer #1: No

Reviewer #2: Yes

Reviewer #3: Yes

Reviewer #1: Dear Authors,

The revised version of your manuscript titled “Molecular mechanism of gallium nitrate in inhibiting bacterial biofilm formation through pykF modulation” presents a significant advancement in the understanding of how gallium nitrate modulates bacterial physiology and biofilm formation. Your integration of transcriptomics, in vitro and in vivo models, and molecular docking is commendable and contributes meaningfully to the field of antimicrobial research.

The inclusion of mechanistic hypotheses about the dual role of pykF under stress conditions—where its upregulation may disrupt bacterial homeostasis and its knockout may trigger compensatory metabolic pathways—is both thoughtful and well-contextualized.

Below are a few points that could further improve the clarity and strength of the manuscript:

Language polishing: Although substantially improved, a few grammatical inconsistencies and awkward phrasings persist (e.g., in the abstract and introduction). A final English editing round would enhance readability.

Justification for strain selection: You have explained why clinical isolates were not used in this study. Still, including a sentence in the Introduction or Discussion stating that this is a preliminary mechanistic study and future work will explore clinical strains would strengthen the rationale.

Mechanism of pykF upregulation: While you provide several plausible mechanisms for the increase in PykF protein without a corresponding mRNA increase in the KO strain, consider briefly stating in the discussion that direct experimental evidence (e.g., proteasomal degradation, ribosomal profiling) is still lacking.

Graphical abstract clarity: The graphical abstract is visually engaging, but it would benefit from labeling key molecules (e.g., Ga³⁺, PykF) and briefly indicating whether their effects are inhibitory or activating.

Minor formatting issues: Please double-check figure legends for consistency and ensure all abbreviations are defined upon first use (e.g., XTT, CLSM).

Overall, the manuscript is well-structured and scientifically sound. I appreciate your thorough responses to the previous review and the additional experiments and discussions incorporated.

Best regards!

Reviewer #2: I thank the authors for responding to the comments, but it was expected that the changes made to the article would be highlighted to allow for their review.

Reviewer #3: (No Response)

**Do you want your identity to be public for this peer review?** For information about this choice, including consent withdrawal, please see our Privacy Policy

Reviewer #1: **Yes:** Amanda Stefanie Jabur de Assis

Reviewer #2: No

Reviewer #3: No

---

## [Author Response · Author response to Decision Letter 2]

21 Aug 2025

Dear reviewers,

We have revised the paper again based on the comments from reviewers and editors. In particular, the Abstract and Conclusion sections have been re-polished, and a certificate of editing has been submitted. Additionally, all points raised by the reviewers requiring revision have been addressed and modified accordingly.

We deeply appreciate your consideration of our manuscript, and we look forward to receiving comments from the reviewers. If you have any queries, please don’t hesitate to contact me at the address below.

Thank you and best regards.

Yours sincerely,

Dr.Junjie Dong

---

## [Editor Report · Decision Letter 2]

27 Aug 2025

Dear Dr. dong,

Thank you for submitting your manuscript to PLOS ONE. After careful consideration, we feel that it has merit but does not fully meet PLOS ONE’s publication criteria as it currently stands. Therefore, we invite you to submit a revised version of the manuscript that addresses the points raised during the review process.

We look forward to receiving your revised manuscript.

Kind regards,

Irene Ling

Academic Editor

PLOS ONE

Journal Requirements:

**Additional Editor Comments:**

We recommend providing clearer elaboration and justifications where experimental evidence is lacking, while also rechecking all figures and abbreviations for consistency and readability to ensure they are easy to follow. Conducting thorough English editing is essential to address any remaining grammatical inconsistencies and improve overall readability.

---

## [Author Response · Author response to Decision Letter 3]

7 Nov 2025

Dear reviewers and editors,

Thank you for the valuable suggestions from the reviewers. We have comprehensively revised the manuscript to ensure it meets the formatting requirements of PLOS ONE. The specific revisions are as follows:

Other issues such as font and formatting have all been revised, and the revisions have been marked in the "Revised Manuscript with Track Changes".

We have submitted the manuscript to a professional institution for language polishing. Issues related to language and grammar in the manuscript have been resolved. Regarding the Statistical Analysis section, relevant annotations have been added in the Results and Discussion parts of the article. The Conclusion section has been rephrased, making it more suitable for publication compared to the previous version.We have once again removed the revision traces and line numbers from the manuscript to ensure compliance with the requirements. Additionally, other issues concerning the Discussion section have also been refined and improved in the manuscript.

We would like to once again express our gratitude to the reviewers for their constructive comments on the discussion section of our manuscript. We sincerely hope that this article can be published in PLOS ONE.

---

## [Editor Report · Decision Letter 3]

11 Nov 2025

Molecular mechanism of gallium nitrate in inhibiting bacterial biofilm formation through pykF modulation

PONE-D-25-24459R3

Dear Dr. dong,

We’re pleased to inform you that your manuscript has been judged scientifically suitable for publication and will be formally accepted for publication once it meets all outstanding technical requirements.

Kind regards,

Irene Ling, PhD

Academic Editor

PLOS ONE
---

## [Editor Report · Acceptance letter]

PONE-D-25-24459R3

PLOS One

Dear Dr. dong,

I'm pleased to inform you that your manuscript has been deemed suitable for publication in PLOS One. Congratulations! Your manuscript is now being handed over to our production team.

Kind regards,

on behalf of

Dr. Irene Ling

Academic Editor

PLOS One